# First direct evidence for direct cell-membrane penetrations of polycationic homopoly(amino acid)s produced by bacteria

Yamato Takeuchi[1,8], Kazunori Ushimaru[1,2,8], Kohei Kaneda[1,8], Chitose Maruyama[1,3,4], Takashi Ito[1,3], Kazuya Yamanaka[5], Yasushi Ogasawara [6], Hajime Katano[1], Yasuo Kato[7], Tohru Dairi [6] & Yoshimitsu Hamano [1,3,4✉]

Bacteria produce polycationic homopoly(amino acid)s, which are characterized by isopeptide backbones. Although the biological significance of polycationic homopoly(amino acid)s remains unclear, increasing attention has recently been focused on their potential use to achieve cellular internalization. Here, for the first time, we provide direct evidence that two representative bacterial polycationic isopeptides, ε-poly-L-α-lysine (ε-PαL) and ε-oligo-L-β-lysine (ε-OβL), were internalized into mammalian cells by direct cell-membrane penetration and then diffused throughout the cytosol. In this study, we used clickable ε-PαL and ε-OβL derivatives carrying a *C*-terminal azide group, which were enzymatically produced and then conjugated with a fluorescent dye to analyze subcellular localization. Interestingly, fluorescent proteins conjugated with the clickable ε-PαL or ε-OβL were also internalized into cells and diffused throughout the cytosol. Notably, a Cre recombinase conjugate with ε-PαL entered cells and mediated the Cre/loxP recombination, and ε-PαL was found to deliver a full-length IgG antibody to the cytosol and nucleus.

[1] Graduate School of Bioscience and Biotechnology, Fukui Prefectural University, Eiheiji-cho, Fukui 910-1195, Japan. [2] Research Institute for Sustainable Chemistry, National Institute of Advanced Industrial Science and Technology (AIST), Tsukuba, Ibaraki 305-8565, Japan. [3] Fukui Bio Incubation Center (FBIC), Fukui Prefectural University, Eiheiji-cho, Fukui 910-1195, Japan. [4] MicrobeChem Inc., Eiheiji-cho, Fukui 910-1195, Japan. [5] Department of Life Science & Technology, Kansai University, Suita, Osaka 564-8680, Japan. [6] Graduate School of Engineering, Hokkaido University, Kita-ku, Sapporo, Hokkaido 060-8628, Japan. [7] Department of Biotechnology, Toyama Prefectural University, Imizu-shi, Toyama 939-0398, Japan. [8] These authors contributed equally: Yamato Takeuchi, Kazunori Ushimaru, Kohei Kaneda. ✉email: hamano@fpu.ac.jp

Homopoly(amino acid)s, which are rare in nature, are produced by bacteria as secondary metabolites (Fig. 1a, b and Supplementary Fig. 1a). To date, six polymers, ε-poly-L-α-lysine (ε-PαL, 1)[1,2], ε-oligo-L-β-lysine (ε-OβL, 2)[3], γ-poly-L/D-diaminobutyric acid[4,5], β-poly-L-diaminopropionic acid[6], ε-poly-L-β-lysine (our recent finding)[7], and γ-poly-L/D-glutamic acid[1,8] have been identified. In addition to these linear polymers characterized by the isopeptide backbones, two branched polymers consisting of repeated dipeptide chains have also been found in bacteria: multi-L-arginyl-poly-L-aspartic acid (also

referred to as cyanophycin)[9,10] and multi-L-diaminopropionyl-poly-L-diaminopropionic acid[11]. Among these eight homopoly(amino acid)s, γ-poly-L/D-glutamic acid is the only example of a naturally occurring polyanionic isopeptide. The others are polycationic, and there is only limited knowledge about the antimicrobial activities of the linear polycationic isopeptides. Although the biological significance of the homopoly(amino acid)s mentioned above has largely eluded researchers, their polyionic properties and polyamide structures are currently an area of focus in biological material science.

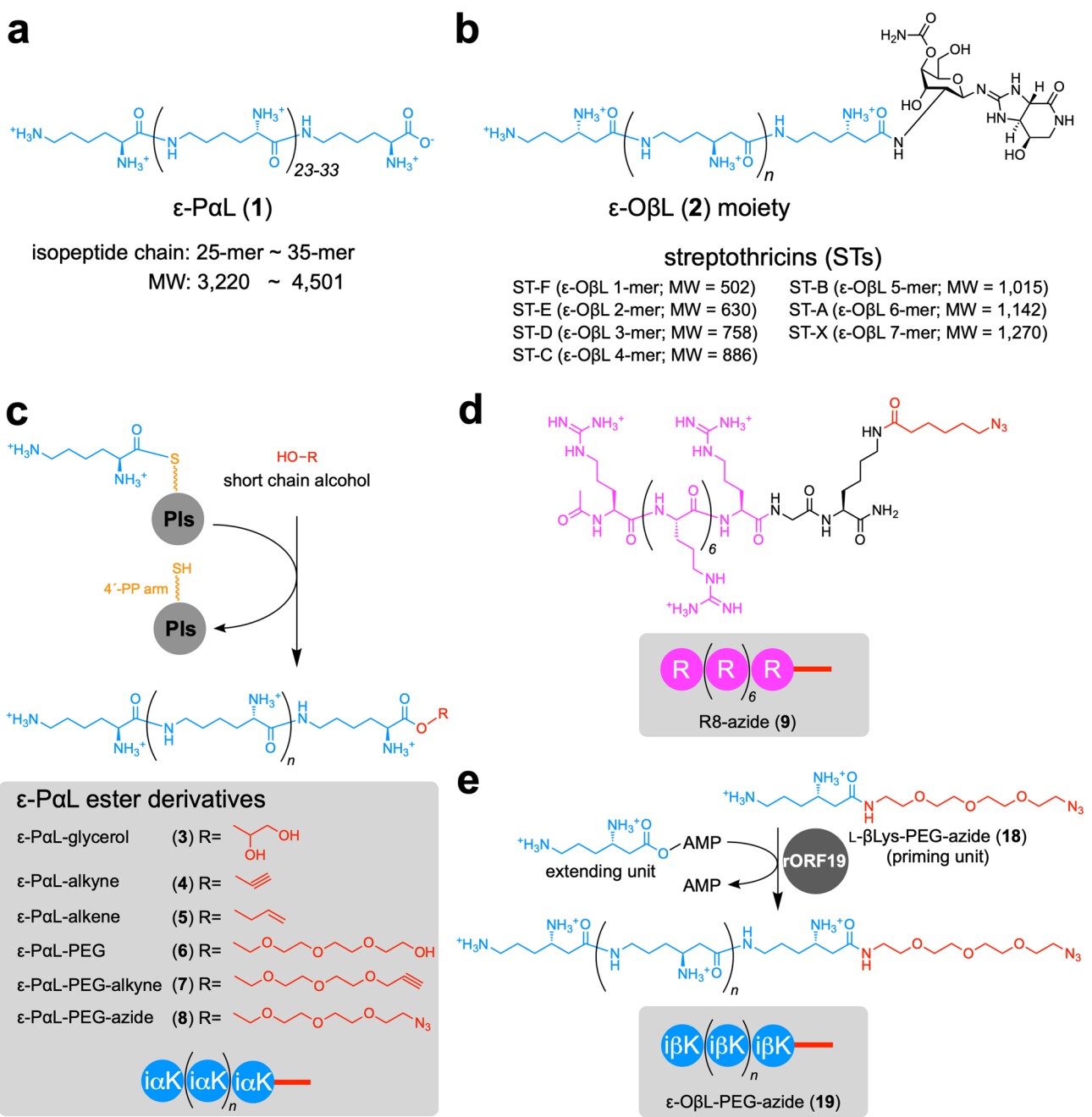

**Fig. 1 Polycationic CPPs used in this study. a, b** Chemical structures of ε-PαL (**1**) and ε-OβL (**2**). Polymer **1** consisting of 25–35 L-αLys residues is produced by *Streptomyces albulus* NBRC14147 as a secondary metabolite (**a**). Oligomer **2** is a substructure of STs. All ST-related compounds consist of carbamoylated D-gulosamine to which **2** (1–7 L-βLys residues) and the amide form of the unusual amino acid streptolidine (streptolidine lactam) are attached. **c** ε-PαL ester derivatives produced by the NBRC14147 strain in this study. Their C-termini were esterified with alcohols, and added to the culture media. Pls catalyzed the polymerization of L-αLys and the esterification reactions. iαK (light blue circles), isopeptide L-αLys monomer unit. **d** The chemical structure of R8-azide (**9**). Peptide **9** was chemically synthesized and used as a canonical CPP control. R (light purple circles), α-peptide L-arginine monomer unit. **e** Enzymatically synthesized ε-OβL-PEG-azide (**19**) by rORF19. iβK (light blue circles), isopeptide L-βLys monomer unit.

Unlike the isopeptides with unknown function, standard peptides (also called eupeptides) with arginine- and/or lysine-rich sequences (typically 5–30 amino acid residues) are currently attracting interest for their cell-penetrating activities in mammalian cells due to their polycationic features at physiological pH; such eupeptides are called cell-penetrating peptides (CPPs)[12–15]. Although amphipathic and hydrophobic CPPs are also known, polycationic CPPs are frequently employed as vehicles to deliver biological macromolecules (cargoes) into mammalian cells[14]. The internalization routes of cationic CPPs are themselves broadly divided into energy-independent direct penetration and energy-dependent endocytosis/macropinocytosis[14,15]; in both routes the CPP bindings to negatively charged cell-membrane components (such as heparan sulfate proteoglycans) are an essential trigger for the internalization events. In contrast to the direct penetration routes, the cargoes taken up by endocytosis/macropinocytosis must escape from endosomes to the cytosol to avoid degradation, reach their molecular targets, and exert their biological activities. Importantly, CPPs carrying a macromolecule cargo such as a protein usually enter cells only by an endocytotic/macropinocytotic route[14–17]. Therefore, recent works have focused on rationally designed synthetic CPPs to obtain more efficient internalizations, both in order to facilitate the direct penetrations or endosomal escapes of the CPP-protein conjugates and to confer resistance to proteolytic degradation[17–21]. In addition to these recent landmark gateways, different approaches to simplify various aspects of the methodology are expected to boost practical intracellular delivery of biological macromolecules. Furthermore, the harmful effects from the polycationic features remain a critical issue to be solved in canonical CPPs with eupeptide structure.

In this context, a bacterium-origin polycationic polymer, ε-PαL (1) (Fig. 1a), has garnered increasing interest in recent years as an alternative to canonical cationic CPPs. Polymer 1 shows potent antimicrobial activities and nondetectable levels of cytotoxicity[1,2], in addition to proteolytic resistance due to the isopeptide-bond linkages between its α-carboxyl groups and ε-amino groups[22]. Recent studies reported that 1 showed potential enhancements of the cellular uptakes of its electrostatic counterparts, i.e., DNA[23–26], siRNA[27], and drug microcapsules[28–31]. However, the cationic charges of 1 were mostly neutralized by the anionic counterparts in their complex forms, raising the question of how 1 carries its anionic cargoes into cells. Moreover, no reports have demonstrated the internalization route of 1 itself, whose molecular weight (approximately 4000) is much higher than those of canonical CPPs.

In addition to 1, the polycationic lysine-isopeptide structures are also found in streptothricin (ST) antibiotics (Fig. 1b); however, their monomer units are not L-α-lysine (L-αLys) but L-β-lysine (L-βLys): namely, ε-OβL (2). STs show antimicrobial activities and cytotoxicity to mammalian cells. Their molecular targets in prokaryotic cells are the ribosomes, in order to inhibit protein biosynthesis, but their cytotoxic mechanism in eukaryotic cells remains unclear. These harmful effects are boosted by the ε-OβL pendant chain (1–7 mer). In fact, STs with a longer ε-OβL pendant chain (ST-B, ST-A, and ST-X) show greater cytotoxicity. Therefore, it has been hypothesized that the ε-OβL chain behaves like a CPP in order to internalize the ST core structure (ST-F) to the cells.

Our previous studies demonstrated that the isopeptide polymer structures in 1 and 2 are nonribosomally synthesized by disparate mechanisms (Supplementary Fig. 1b, c)[2,3]. In this study, we generated clickable derivatives of 1 and 2 by utilizing their biosynthetic enzymes with the relaxed substrate specificities in the polymerization reactions. We then clarified their cell-penetrating activities using conjugates with a fluorescent dye produced by click chemistry. This report further describes the outstanding internalizations of 1-protein and 2-protein conjugates into living mammalian cells. We also demonstrate the ability of ε-PαL decoration to deliver a full-length IgG antibody (150 kDa) into cells, which could dramatically transform therapeutic approaches and foster experimental breakthroughs in molecular biology.

## Results

**Microbiological production of clickable ε-PαL ester derivatives.** Installation of a clickable functional group at the C-terminus of 1 is in demand to decorate materials without any additional electrostatic counterparts; this approach does not eliminate the polycationic feature of 1. However, there have been quite a few successful chemical approaches to solve this issue. In contrast, it has been reported that an ε-PαL-producing bacterium, *Streptomyces albulus* NBRC14147, can produce ε-PαL ester derivatives when the culture medium is supplemented with short-chain polyols[32]. Our present study also confirmed the production of an ester derivative in the culture medium containing 0.2% (w/v) glycerol; the C-terminus of 1 was esterified with glycerol, producing ε-PαL-glycerol (3) (Fig. 1c, Supplementary Fig. 2a, b, and Supplementary Tables 1, 2). The isopeptide chain-shortening of 3 was a common observation during the esterification with short-chain polyols[32].

The esterification mechanism remains uncertain. However, these findings allowed us to examine whether a reactive group, such as an alkyne, alkene, or azide group, could be installed at the C-terminus of 1 during the esterification process. We first examined whether 2-propyn-1-ol and 3-buten-1-ol could be incorporated into the C-terminus. For this purpose, the NBRC14147 strain was cultured in a medium supplemented with 2-propyn-1-ol or 3-buten-1-ol. The culture broth was analyzed by high-performance liquid chromatography and high-resolution electrospray ionization mass spectrometry (HPLC-HR-ESI-MS). As expected, 0.2% (w/v) supplements of these two alcohols resulted in the production of the corresponding ester derivatives (Supplementary Fig. 2c, d and Supplementary Tables 3, 4). Similarly, three other alcohols, polyethylene glycol (PEG), triethylene glycol mono(2-propynyl) ether (PEG-alkyne), and 11-azido-3,6,9-trioxaundecanol (PEG-azide), were also incorporated into the C-terminus of 1 via an ester linkage (Supplementary Fig. 2e–g and Supplementary Tables 5–7). These esterification reactions occurred in around 70% of ε-PαL produced (Supplementary Table 8). We further showed that, in vitro, the recombinant ε-PαL synthetase (rPls) produced two ester derivatives, ε-PαL-glycerol (3) and ε-PαL-PEG (6), in the reaction mixture containing glycerol and PEG, respectively (Supplementary Fig. 3 and Supplementary Tables 9, 10). These findings demonstrated that rPls directly catalyzed the esterification reactions.

Among the ε-PαL ester derivatives, ε-PαL-PEG-azide (8), was purified from the culture broth and used for further experiments after confirming its chemical structure by NMR analysis (Supplementary Table 11).

**Cellular uptake of ε-PαL (1) in HeLa cells.** To fluorescently label 1, compound 8 was covalently attached to a fluorescein (FAM, 0.4 kDa) derivative carrying a dibenzocyclooctyne (DBCO) group by click reaction (copper-free azide-alkyne cycloaddition) (Supplementary Table 12). The resulting compound ε-PαL-FAM (10) (Fig. 2) was then employed to investigate the cell-penetrating activity of 1.

Confocal microscopy (CM) was initially employed to assess the cellular internalization of 10 (Fig. 3a). HeLa cells were incubated

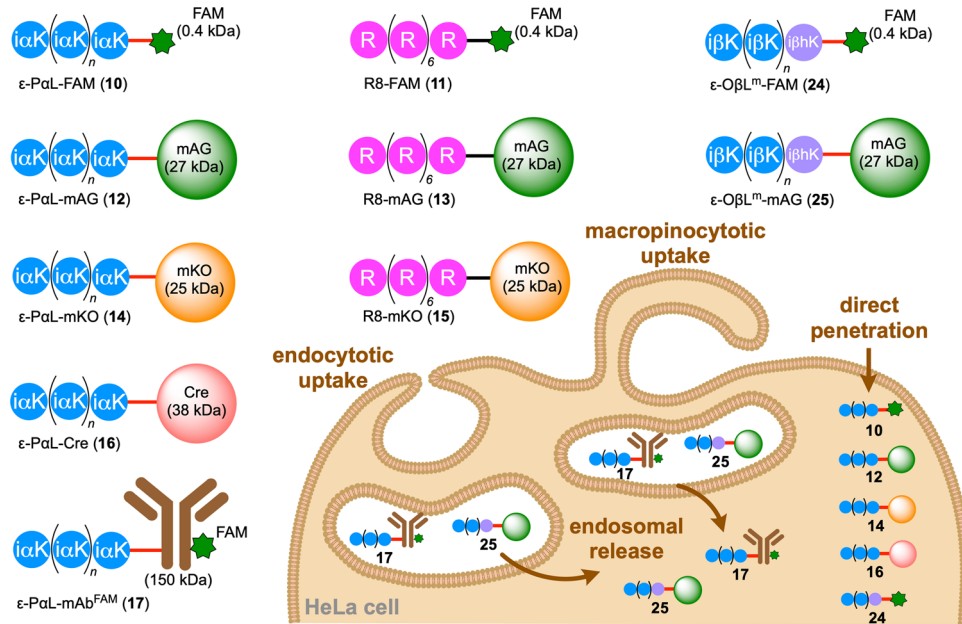

**Fig. 2 Schematic diagrams of cargoes conjugated with polycationic peptides.** A fluorescein dye (FAM) and proteins (mAG, mKO, and mAb) were conjugated with ε-PαL, R8, or ε-OβL^m by click chemistry, and the resulting products are schematically shown. Their chemical structures validated by HPLC-HR-ESI-MS data are shown in the Supplementary Tables. Our present study demonstrated the dominant pathways for their intracellular uptake (bottom right). Compounds **10**, **12**, **14**, **16**, and **24** were internalized into cells mainly by direct penetration. However, **17** and **25** were taken up into cells by endocytosis and/or macropinocytosis and were then released from endosomes. Compounds **11**, **13**, and **15** were used as canonical CPP controls.

with **10** (50 μM) under typical cell culture conditions with serum. Before imaging, cells were washed with heparin, which is highly negatively charged and should strip **10** electrostatically associated with the plasma membrane surface. The cells were then fixed with 4% paraformaldehyde[14]. In the incubation at 37 °C for 60 min, CM demonstrated that **10** was internalized into cells and diffused throughout the cytosol. The observed green-fluorescent image indicated the direct translocation route for **10** because there were no punctate signals suggestive of endosomes. Moreover, diffuse signals of **10** were also observed on intracellular uptake at 4 °C, a temperature at which energy-dependent endocytosis was strongly suppressed. These findings revealed that the cellular internalization of **10** occurred through a direct penetration pathway. However, a low concentration (2 μM) of **10** gave only punctate signals suggestive of endosomes at 37 °C (Fig. 3b). These dose-dependent signals (diffuse or punctate) have been reported in canonical synthetic CPPs, such as octa-arginine (R8) and dodeca-arginine (R12)[33,34]. We indeed observed such signals in the treatment with R8-FAM (**11**) in hand (Fig. 2 and Fig. 3a, b). Therefore, like R8, **1** entered the cells by two pathways: direct penetration and endocytosis/macropinocytosis. For this internalization, the covalent-bond conjugation was needed; a non-covalent approach resulted in no cellular uptake (Fig. 3c). Serum in the culture medium did not affect the cellular internalization ratio (Fig. 3c), but polyanionic heparin significantly inhibited the cell-penetrating event (Fig. 3d), strongly suggesting that negatively charged membrane components, including heparan sulfate proteoglycan, are essential for the ε-PαL internalization. Moreover, quick cellular uptakes (within 5–15 min) and dose-dependent uptakes of **10** occurred at 4 °C and 37 °C (Fig. 3e–h). PAO (an endocytosis inhibitor)[35] and EIPA (a macropinocytosis inhibitor)[36] suppressed these cellular uptakes (Fig. 3i). However, the levels of inhibition were lower than that by **11** (Fig. 3j), suggesting that the main pathway for the ε-PαL internalization is direct penetration across the cell membranes. We therefore turned our attention to the cytosolic delivery of protein cargoes by **1**.

**Cytosolic deliveries of protein cargoes conjugated with ε-PαL (1).** A 27 kDa green-fluorescent protein, monomeric Azami-Green (mAG), was chemically derivatized with DBCO-Sulfo-N-hydroxysuccinimidyl ester to introduce a DBCO group on the mAG surface and subsequently conjugated with **8** by click reaction, yielding ε-PαL-mAG (**12**) (Fig. 2, Supplementary Table 13). HeLa cells were incubated with **12** (50 μM) for 60 min at 37 °C or 4 °C, and then washed and fixed as described above. When the cells were incubated at 37 °C, CM analysis revealed diffuse and vesicular signals (Fig. 4a). Incubation at 4 °C resulted in condensed fluorescent signals within nuclei and diffused green-fluorescent signals in cytosols (Fig. 4a). While the **12** covalent-bond conjugate could deliver the protein cargo into cells, a noncovalent approach resulted in no cellular uptake (Fig. 4b). Despite the large cargo (27 kDa), **12** entered the cells quickly within 5–15 min, and its dose-dependent uptakes were also shown at both 4 °C and 37 °C (Fig. 4c–f). Endocytosis/macropinocytosis inhibitors did not abolish the cellular internalization (Fig. 4g). Unlike **12**, mAG conjugated with R8 (R8-mAG, **13**) (Fig. 2, Supplementary Table 14) was hardly internalized into cells at 4 °C and was trapped in endosomes at 37 °C (Fig. 4a).

To confirm the direct penetration event in **12**, we performed time-lapse imaging with CM. Notably, **12** showed the formation of fluorescence-enriched spots on the cell membranes in the initial internalization phase (Fig. 5a) (Supplementary Video 1). From the spots, **12** then gradually spread throughout the cytosol and nucleus. Such spots were also observed during the direct penetration process of canonical arginine-rich peptides and were previously described as nucleation zones (NZs)[14,37,38]. ε-PαL (isopeptide) and arginine-rich peptides (eupeptide) should be structurally classified into different classes, but interestingly, ε-PαL can deliver protein cargoes directly and very quickly into cells through NZ. Three direct penetration mechanisms have been proposed in CPPs having α-helix conformations (Fig. 5a)[15]. ε-PαL has been suggested to form a β-sheet conformation[39,40]. Although the intracellular uptake mechanism remains unclear,

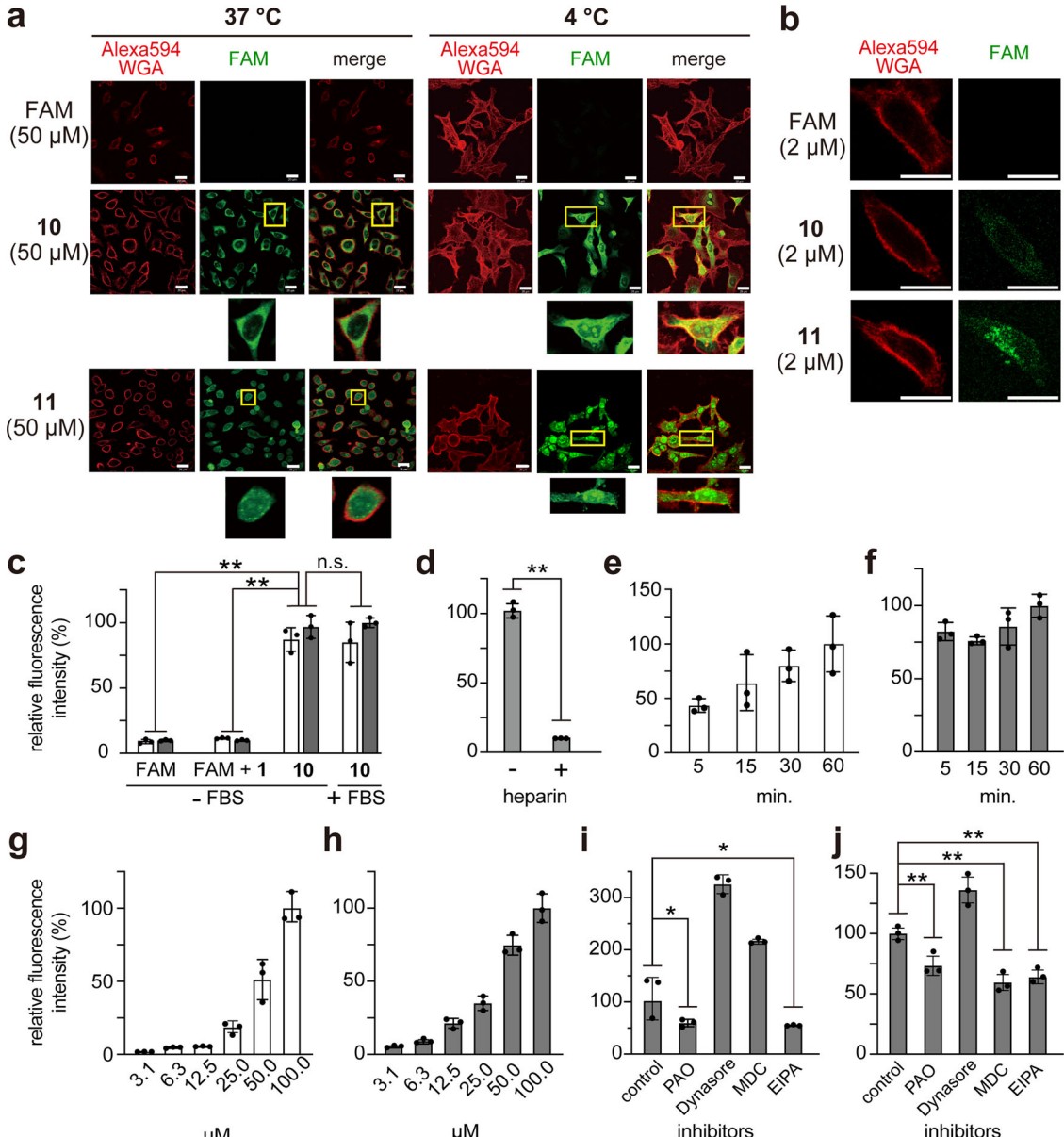

**Fig. 3 Cellular uptakes of ε-PαL-FAM (10). a, b** Cytosolic delivery of FAM conjugated with ε-PαL (**1**) or R8. HeLa cells were incubated with FAM, ε-PαL-FAM (**10**), and R8-FAM (**11**) under typical cell culture conditions with serum. After washing and fixing the cells, the cellular localization (green) of FAM, **10**, and **11** were determined by confocal microscopy (CM). The cell membrane was stained with the membrane marker wheat germ agglutinin (Alexa594-WGA; red). The representative CM images and the magnified views of the yellow-boxed areas are shown. Scale bars, 20 μm. Cells were incubated with 50 μM each of FAM, **10**, and **11** for 60 min at 37 °C and 4 °C (**a**). Cells were incubated with 2 μM each for 60 min at 37 °C (**b**). **c–j** Relative cellular uptakes of **10** at 37 °C (light gray) and 4 °C (white). FAM (50 μM), FAM blended with **1** (FAM + **1**, 50 μM each), and **10** (50 μM) were incubated with cells in the culture medium supplemented with or without serum (FBS) for 60 min (**c**). The results are presented as the mean ± standard deviation (s.d.) ($n = 3$). **Significant at $p < 0.01$ by two-way ANOVA followed by Sidak's multiple comparisons test. n.s. not significant. Compound **10** was incubated with cells in the culture medium supplemented with or without mg ml$^{-1}$ heparin for 60 min at 37 °C (**d**). The results are presented as the mean ± s.d. ($n = 3$) **Significant at $p < 0.01$ by unpaired Student's $t$-test. Compound **10** was incubated with cells for 5–60 min at 4 °C (**e**) and 37 °C (**f**). Compound **10** (3.1–100 μM) was incubated with cells for 60 min at 4 °C (**g**) and 37 °C (**h**). Compounds **10** (**i**) and **11** (**j**) were incubated with cells for 60 min at 37 °C in the culture medium supplemented with an endocytosis/macropinocytosis inhibitor (PAO, Dynasore, MDC, or EIPA). The results are presented as the mean ± s.d. ($n = 3$). Asterisks indicate significance at *$p < 0.05$, **$p < 0.01$ by one-way ANOVA followed by Tukey's multiple comparisons test.

the cellular uptakes and cytosolic distributions of **12** were observed in all cell lines tested (Supplementary Fig. 4). To examine the versatility of the ε-PαL vehicle for cytosolic delivery, an alternative fluorescent protein, monomeric Kusabira-Orange (mKO, 25 kDa), was conjugated with **8**, yielding ε-PαL-mKO (**14**) (Fig. 2 and Supplementary Table 15). As expected, identical results were obtained with **14** (Supplementary Fig. 5). mKO conjugated with R8 (R8-mKO) (**15**) (Fig. 2 and Supplementary Table 16) was only taken up again when incubated at 37 °C. Furthermore, time-lapse imaging with CM confirmed that cellular internalization occurred only by the endocytic pathway (Supplementary Video 2). These findings showed that poly-cationic modification by ε-PαL is quite effective for the cytosolic delivery of protein cargoes.

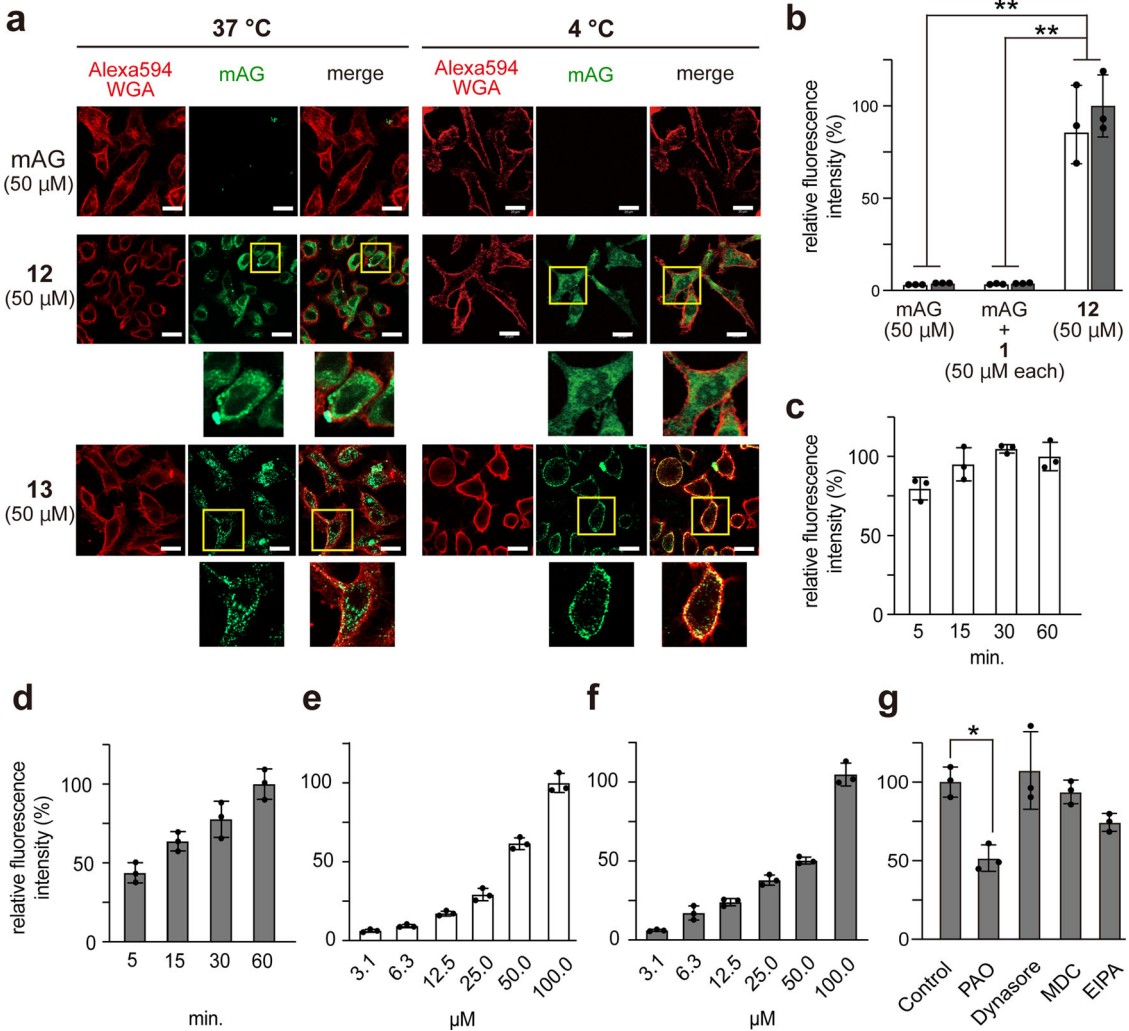

**Fig. 4 Cytosolic deliveries of mAG conjugated with ε-PαL. a** Cytosolic delivery of mAG conjugated with ε-PαL (**1**) or R8. HeLa cells were incubated with mAG, ε-PαL-mAG (**12**), and R8-mAG (**13**) for 60 min at 37 °C and 4 °C. After washing and fixing the cells, the cellular localization of **12** (green) and **13** (green) was determined by CM. The cell membrane was stained with Alexa594-WGA (red). The representative CM images and the magnified views of the yellow-boxed areas are shown. Scale bars, 20 µm. **b–g** Relative cellular uptakes of **12** at 37 °C (light gray) and 4 °C (white). mAG, mAG blended with **1** (mAG + **1**) and **12** were incubated with HeLa cells for 60 min at 37 °C and 4 °C. The results are presented as the mean ± s.d. ($n = 3$). **Significant at $p < 0.01$ by two-way ANOVA followed by Sidak's multiple comparisons test (**b**). Protein **12** (50 µM) was incubated with cells for 5–60 min at 4 °C (**c**) and 37 °C (**d**). Protein **12** (3.1–100 µM) was incubated with cells for 60 min at 4 °C (**e**) and 37 °C (**f**). Effect of endocytosis/macropinocytosis inhibitors on the cellular uptake of **12**. HeLa cells were incubated with **12** for 60 min at 37 °C in the culture medium supplemented with the inhibitor (PAO, Dynasore, MDC, or EIPA). The results are presented as the mean ± s.d. ($n = 3$). *Significant at $p < 0.05$ by one-way ANOVA followed by Tukey's multiple comparisons test (**g**).

Tracking CPP-protein conjugates after their intracellular uptake is important for evaluating their capacities as CPPs. We therefore investigated the dynamic distribution of the intracellular **12**. Time-lapse imaging with optical sectioning microscopy demonstrated that the diffused signal of **12** was continuously detected for at least 12 h (Supplementary Video 3). In contrast, the R8-mKO (**15**) signals were continuously detected as vesicular forms for up to 12 h (Supplementary Video 4). Thus, most of the **15** molecules were constantly trapped in endosomes.

To examine whether the enzyme cargoes delivered by ε-PαL exhibit their functions within cells, we prepared Cre recombinase (38 kDa) conjugated with **8** (ε-PαL-Cre, **16**) (Fig. 2 and Supplementary Table 17). We first transfected HEK293T cells with a Cre activity reporter plasmid (pmKO_loxP_mAG) that expresses an mKO-mAG fusion protein and mKO alone with and without **16**, respectively. We then treated the transfected cells with 50 µM of **16**, and microscopically monitored the reporter

gene expression. As expected, addition of **16** resulted in the expression of mKO-mAG, demonstrating the successful delivery and action (Cre/loxP recombination) of **16** (Fig. 5b).

The successes of ε-PαL-driven intracellular deliveries of mAG (27 kDa), mKO (25 kDa), and Cre recombinase (38 kDa) turned our attention to the delivery of a full-length IgG antibody, which is still challenging due to its high molecular weight (150 kDa)[41]. To examine the capability of ε-PαL for antibody internalization, a fluorescently labeled monoclonal anti-α-tubulin antibody (mAb-FAM) was conjugated with **8**, yielding ε-PαL-mAb^FAM (**17**) (Fig. 2 and Supplementary Table 18). We speculated that a high concentration treatment (50 µM) could not give us a fluorescent image of the microfilamentous cytoskeleton due to excess fluorescent signals. In contrast, the low concentration (e.g., ~2 µM) would result in only punctate signals suggestive of endosomes. Therefore, HeLa cells were incubated with 10 µM of **17**. After the incubation for 2 h at 37 °C and 4 °C, CM analysis

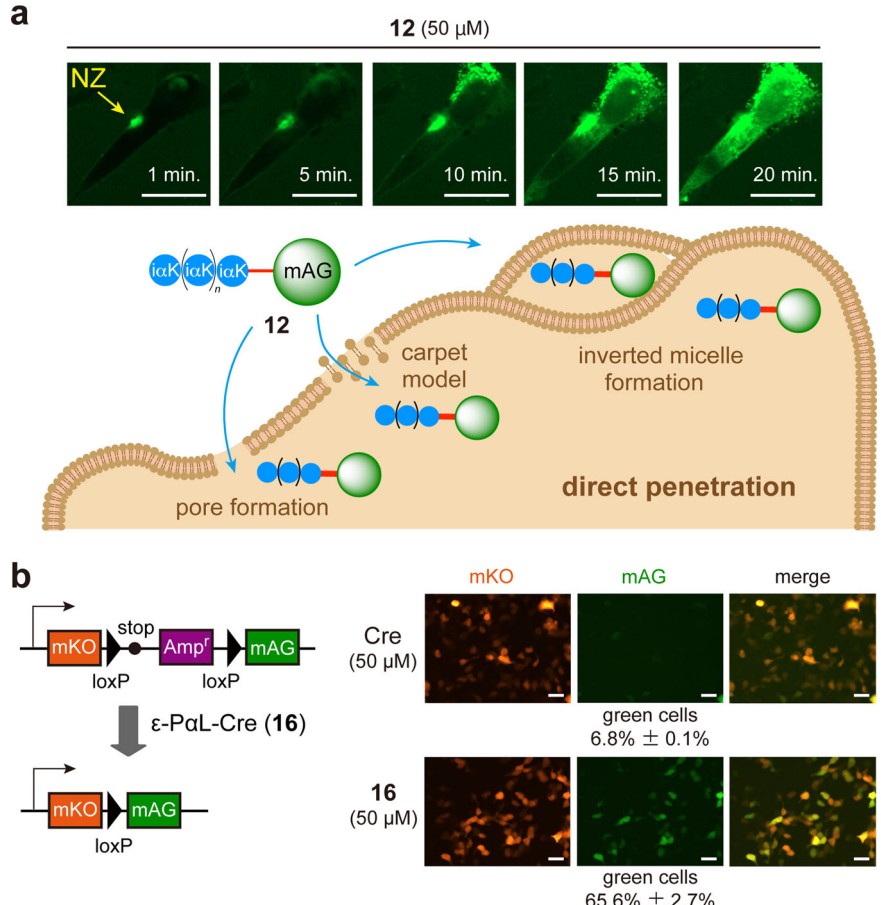

**Fig. 5 Cytosolic deliveries of protein cargoes conjugated with ε-PαL. a** Time-lapse experiment showing the cellular uptake of **12**. The time-lapse imaging showed HeLa cells with CM after the addition of **12** at 37 °C. After 1, 5, 10, 15, and 20 min, the snapshot images are shown (see also Supplementary Video 1). NZ, nucleation zone. Scale bars, 20 μm. The proposed direct penetration mechanisms are schematically shown (bottom). **b** The Cre activity reporter plasmid constructed in this study is schematically shown. The reporter plasmid expressed a fusion fluorescent protein (mKO-mAG) and mKO alone in HeLa cells with and without Cre recombinase, respectively. HEK293T cells were transfected with the reporter plasmid and were then incubated for 24 h at 37 °C. The cells were treated with Cre recombinase (50 μM) or ε-PαL-Cre (**16**) (50 μM) for 60 min at 37 °C after confirming the mKO expression. Cells were incubated for 24 h under typical culture conditions after washing. The gene recombination event by **16** was monitored by microscopy. The representative microscopy images are shown. The gene recombination ratios are presented as the mean ± s.d. ($n = 3$). Scale bars, 20 μm.

revealed both diffusing and punctate signals of **17** in the cells incubated at 37 °C, but not those incubated at 4 °C (Fig. 6a). These findings strongly suggested that **17** taken up mainly by endocytosis and/or macropinocytosis was released from the endosomes and distributed in the cell. More surprisingly, the condensed fluorescent signals were also detected within nuclei. Conversely, **17** did not provide a fluorescent image of the microfilamentous cytoskeleton, although **17** did so in a standard immunostaining experiment (Fig. 6b). The cytosolically diffused signals of **17** probably resulted from the excess and α-tubulin-unbound **17** that is unremovable in the living cells.

**Cytotoxicity of ε-PαL (1).** To address the growing worldwide demand for CPPs, structurally and functionally distinct CPPs are required to circumvent the canonical CPP drawbacks such as cytotoxicity. This study, therefore, focused on the cytotoxicities of ε-PαL (**1**) against mammalian cells, while the nontoxicity of **1** in vivo has also been reported so that ε-PαL is used as a natural food preservative in several countries[1,42]. However, the in vitro cytotoxicity of **1** in other cell lines has remained unclear. We therefore performed the MTT cell viability assay with **1** in K562, HepG2, and HEK293T cells in addition to HeLa cells, and we

confirmed that **1** did not show cytotoxicities in these cell lines (Supplementary Table 19).

**Chemoenzymatic synthesis of a clickable ε-OβL derivative with an azide group.** As mentioned earlier, we hypothesized that the ε-OβL moiety (**2**) behaves like a CPP to internalize the ST core structure (ST-F) within cells (Fig. 1b). We therefore investigated the cell-penetrating activity of **2** in mammalian cells. We employed a chemoenzymatic approach to introduce an azide group at the C-terminus of **2**.

In the ST biosynthesis, the ε-OβL chain growth (2–7 mer) is mediated by ORF19, a stand-alone adenylation domain of nonribosomal peptide synthetase (NRPS) (Supplementary Fig. 1c)[3]. In this chain-elongation reaction, ORF19 never oligomerizes directly with only L-βLys monomer units; rather, ORF19 requires the L-βLys scaffold tethered to 4′PP-ORF18 as a priming unit[3]. Aminoacyl-N-acetylcysteamine thioesters (aminoacyl-SNACs) are often used as mimicking substrates of the aminoacyl substrate tethered to the 4′-PP arm[43]. However, instead of the aminoacyl-SNACs, we examined whether L-βLys tethered to a chemically synthesized PEG-azide tail (L-βLys-PEG-azide, **18**) (Fig. 1e and Supplementary Table 20) would be acceptable as a priming unit for recombinant ORF19 (rORF19).

## a

**17** (10 μM)

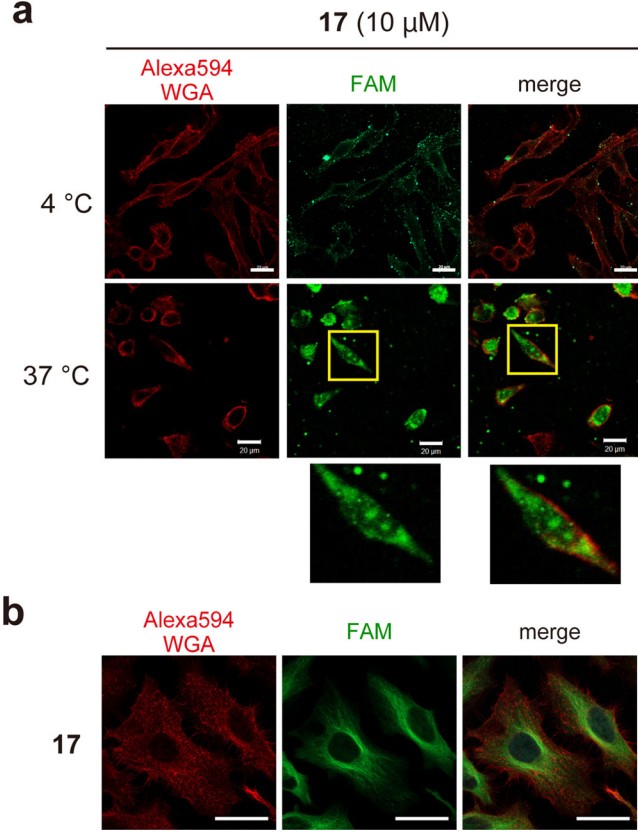

## b

**Fig. 6 Cytosolic delivery of IgG conjugated with ε-PαL. a** HeLa cells were incubated with ε-PαL-mAb[FAM] (**17**) for 120 min at 37 °C and 4 °C. After washing and fixing the cells, the cellular localization of **17** (green) was determined by CM. The cell membrane was stained with Alexa594-WGA (red). Representative CM images and magnified views of the yellow-boxed areas are shown. Scale bars, 20 μm. **b** immunostaining of α-tubulin by **17** in HeLa cells. The cells were fixed with 4% paraformaldehyde, permeabilized in 0.1% Triton X-100 in PBS, then blocked in a blocking reagent (Blocking One). The cells were then incubated with **17** at 2 μg ml⁻¹ (shown in green) for 60 min at 25 °C. The cell membrane was stained with Alexa594-WGA (red) after washing the cells, and the cells were then observed by CM. Scale bars, 20 μm.

We incubated compound **18** (priming unit) and ʟ-βLys (extending unit) with rORF19 (0.1 mg ml⁻¹) and ATP. Fortunately, the HPLC-ESI-HR-TOF analysis of the reaction mixture showed that rORF19 mediated an ATP-dependent chain elongation (2–4 mer) on the **18** scaffold and produced ε-OβL-PEG-azide (**19**) (Fig. 1e, Supplementary Fig. 6a, b and Supplementary Table 21). This finding demonstrated that **18** is acceptable as a priming unit. We also examined three different priming units (Supplementary Fig. 6a) based on the substrate specificity of ORF19 previously reported by our group[3]. An ʟ-βLys analog, ʟ-β-homolysine (ʟ-βhLys), was previously acceptable. Therefore, ʟ-βhLys-PEG-azide (**20**) was chemically synthesized (Supplementary Table 22) and used in the rORF19 reaction. As expected, rORF19 mediated the chain growth of ʟ-βLys isopeptides in the presence of **20** (Supplementary Fig. 6c and Supplementary Table 23). Priming unit **20** was more effective than **18** based on two findings: (i) the productivities of the elongated isopeptides were improved, and (ii) compound **20** was more consumed by rORF19. Conversely, ʟ-αLys-PEG-azide (**22**) (Supplementary Table 24) and NH₂-PEG-azide (**23**) were not accepted as priming units (Supplementary Fig. 6d, e). From these findings, we selected **20** as the suitable priming unit and designated the enzymatically

produced oligomer as ε-OβLᵐ-PEG-azide (**21**) (Supplementary Fig. 6f).

By increasing the rORF19 concentration from 0.1 to 1.0 mg ml⁻¹, the reaction mixture produced **21** with longer isopeptide chains (2–9 mer) (Supplementary Table 25). The resulting oligomer was purified from the reaction mixture and then used again as the priming unit. This 2nd round enzyme reaction yielded oligomers that were more extended (2–13 mer) (Supplementary Table 25), which were used as clickable ε-OβLs for further experiments.

**Cell-penetrating activity of ε-OβL (2).** Compound **21** had chains of much shorter lengths than **8** (Supplementary Table 7). We therefore first investigated the relationship between the cell-penetrating activity and peptide chain length. The **21** oligomers were purified according to chain length (2, 3, 4, 5, 6, and 7–13 mer). The six oligomers with different chain lengths, **21** (2 mer), **21** (3 mer), **21** (4 mer), **21** (5 mer), **21** (6 mer), and **21** (7–13 mer), were then respectively conjugated with FAM by click reaction, producing ε-OβLᵐ-FAM (**24**) (2 mer), **24** (3 mer), **24** (4 mer), **24** (5 mer), **24** (6 mer), and **24** (7–13 mer) (Supplementary Table 26). HeLa cells were incubated with **24** (50 μM) at 37 °C and 4 °C for 60 min. CM analysis demonstrated no cell-penetrating activities in the short chains (2 mer and 3 mer) (Fig. 7). In contrast, the longer chains (>4 mer) entered cells at 37 °C and 4 °C and arrived at the nucleus. For quantitative analysis, we prepared **24** consisting only of the cell-permeable isopeptides (4–13 mer) (Supplementary Table 27), and it was significantly, quickly, and dose-dependently internalized into cells at 37 °C and 4 °C (Supplementary Fig. 7a–e). During incubation with endocytosis/macropinocytosis inhibitors, the intracellular uptake of **24** (4–13 mer) was suppressed only by EIPA (Supplementary Fig. 7f). These findings revealed that internalization of **2** was performed by direct penetration and macropinocytosis. More interestingly, these results implied that **2**, a substructure of STs, functions as a CPP to deliver the ST core structure (ST-F) into cells.

**Cytosolic delivery of a fluorescent protein decorated with ε-OβL.** We conjugated mAG with **21** (4–13 mer) by click reaction to examine whether ε-OβL can deliver macromolecules intracellularly (Supplementary Table 28). The resulting conjugate, ε-OβLᵐ-mAG (**25**) (Fig. 2), was added to HeLa cell culture at 50 μM and incubated for 60 min at 37 °C or 4 °C. Interestingly, both the diffusing and vesicular signals of **25** were observed within cells under incubation at 37 °C (Fig. 8a). However, under incubation at 4 °C, **25** was trapped by the cell membrane and hardly internalized into cells (Fig. 8a, b). These data showed that **25** was internalized mainly by endocytosis/macropinocytosis. The following results also support this: (i) intracellular delivery was significantly suppressed by PAO and EIPA (Fig. 8c), and (ii) **25** needed slightly more time (~60 min) (Fig. 8d) than **12** to enter the cells, while dose-dependent uptakes of **25** occurred at 37 °C (Fig. 8e). Conversely, we simultaneously observed the diffusing signals in the **25**-treated cells at 37 °C (Fig. 8a), demonstrating that **25** was released from endosomes and then diffused to the cytosol (Fig. 2).

## Discussion

Our explorations showed that the cellular uptakes of **1** and **2**, which had a small cargo (e.g., FAM in this study), occurred mainly by energy-independent direct penetration. Surprisingly, direct penetrations were also observed in **12** (27 kDa) and **14** (25 kDa), while the canonical CPP (R8) mediated the deliveries of protein cargoes only via endocytosis/macropinocytosis. Considering that **16** (38 kDa) mediated its Cre/loxP recombination in

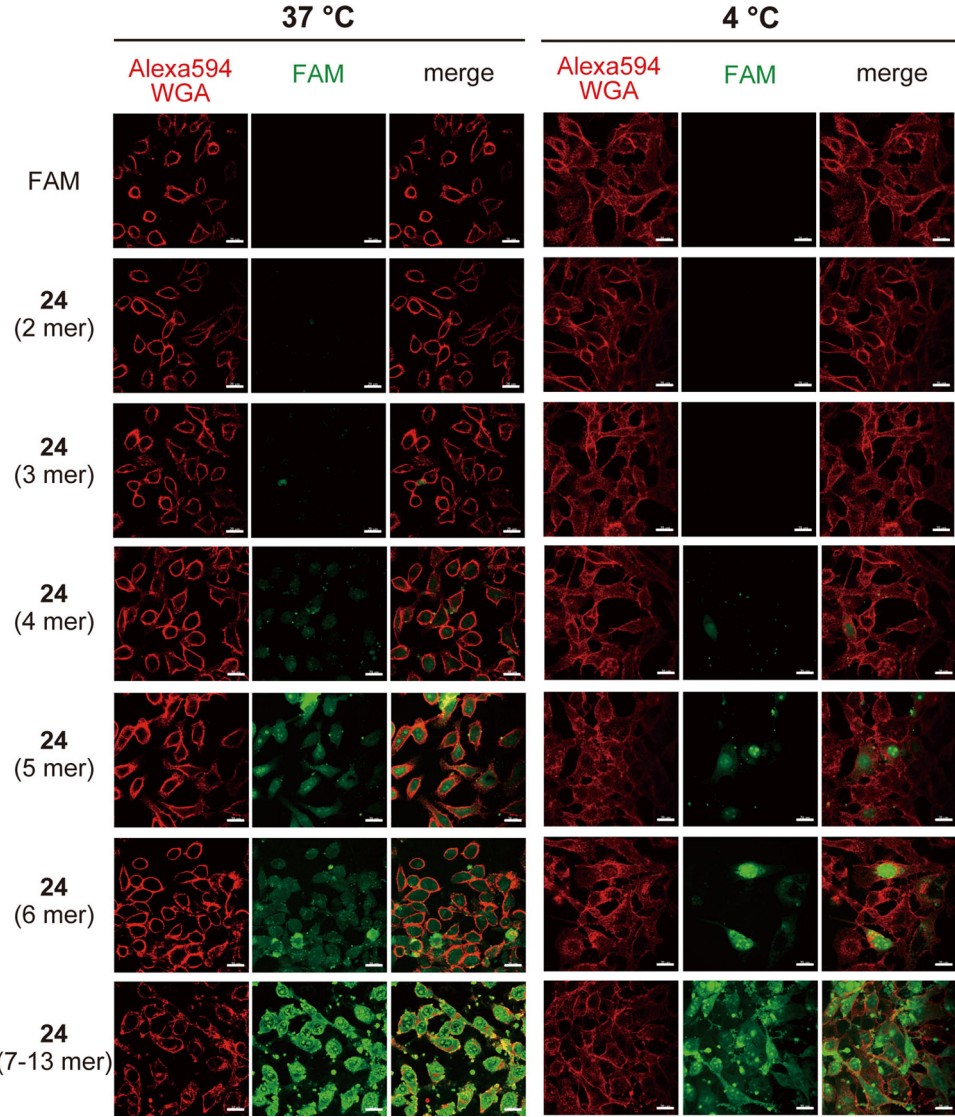

**Fig. 7 Cell-penetrating activity of ε-OβL.** HeLa cells were incubated with 50 μM of ε-OβL$^m$-FAM (**24**) (2 mer), **24** (3 mer), **24** (4 mer), **24** (5 mer), **24** (6 mer), or **24** (7–13 mer) for 60 min at 37 °C or 4 °C under a typical cell culture condition with serum. After washing and fixing the cells, the cellular localization (green) of **24** was determined by CM. The cell membrane was stained with Alexa594-WGA (red). The representative CM images are shown. Scale bars, 20 μm.

the nucleus and that **17** (150 kDa) escaped from endosomes and then arrived at the nucleus, we conclude that the polycationic conjugation approach with **1** is effective for both the nucleus-targeted delivery and the cytosolic delivery of protein cargoes. Our present study also revealed that the ε-OβL chain of antibiotic STs functions as a CPP to deliver the ST core structure (ST-F) into cells. Furthermore, to our surprise, we found that the conjugation with **2** facilitated endosomal escape of protein cargoes. Although we employed **21** (4–13 mer) in this study, elongation of the isopeptide-chain length could enable direct penetration of protein cargoes.

Practical intracellular delivery of biological macromolecules (biopharmaceuticals) would dramatically transform therapeutic approaches and foster experimental breakthroughs because numerous potential drug targets (cytokines, growth factors, and signal transduction proteins) are in the cell interior[44]. CPPs are among the most promising strategies, and CPP-driven technology currently revolves around natural and synthetic eupeptides. Poly-L/D-arginine[13–15], poly-L-histidine[45], and poly-L/D-lysine[46], which were chemically or genetically produced, have been used as CPPs.

To enable the delivery of protein cargoes by these eupeptides, recombinant expression of CPP-cargo fusions at the N- or C-terminus is a straightforward strategy. However, low-level expressions in a host strain are known problems due to the insolubility, toxicity, and truncation of the fusion proteins[16,47]. In contrast, the chemical in vitro attachment of a CPP to a protein surface is a promising alternative approach. This bioconjugation process relies on the chemical synthesis of CPPs that often contain a reactive group-end specific to functional groups (e.g., primary amine and sulfhydryl groups) on the protein surface. However, the resulting CPP sway arm on the protein body is often biochemically unstable due to proteolytic degradation[48]. Although the cyclic CPPs (e.g., cyclic Tat peptides and cyclic oligoarginines) have great potential for overcoming the critical degradation problem[20,48,49], a much more versatile and facile method without laborious synthesis would expand the application potential of the CPP-driven technology. Therefore, the naturally occurring polycationic isopeptides, **1** and **2**, which are resistant to peptidases/proteinases[22] and are nontoxic to mammalian cells[1–3], are a promising new class of CPPs. Moreover, their biosynthetic

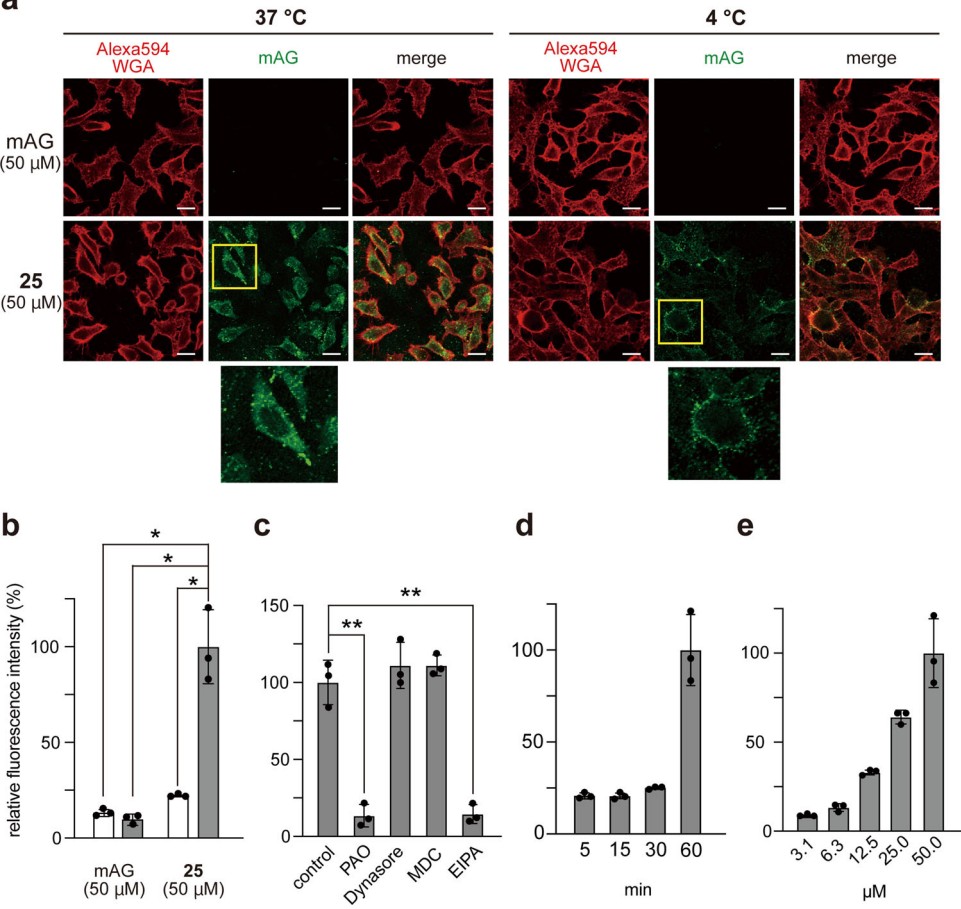

**Fig. 8 Cytosolic deliveries of mAG conjugated with ε-OβL. a** Cytosolic deliveries of mAG conjugated with ε-OβL (**2**). HeLa cells were incubated with mAG and ε-OβL-mAG (**25**) for 60 min at 37 °C and 4 °C. The cellular localization of **25** (green) was determined by CM after washing and fixing the cells. The cell membrane was stained with Alexa594-WGA (red). The representative CM images and the magnified views of the yellow-boxed areas are shown. Scale bars, 20 μm. **b**–**e** Relative cellular uptakes of **25** into HeLa cells. HeLa cells were incubated with 50 μM (unless otherwise noted) of mAG or **25**. mAG and **25** were incubated with cells for 60 min at 37 °C (light gray) and 4 °C (white) (**b**). The results are presented as the mean ± s.d. ($n = 3$). *Significant at $p < 0.05$ by two-way ANOVA followed by Sidak's multiple comparisons test. Protein **25** was incubated with cells for 60 min at 37 °C in the culture medium supplemented with an endocytosis/macropinocytosis inhibitor (PAO, Dynasore, MDC, or EIPA) (**c**). The results are presented as the mean ± s.d. ($n = 3$). **Significant at $p < 0.01$ by one-way ANOVA followed by Tukey's multiple comparisons test. Protein **25** was incubated with cells for 5–60 min at 37 °C (**d**), and **25** (3.1–50 μM) was incubated with cells for 60 min at 37 °C (**e**).

machineries were found to be useful alternatives for installing a clickable group at the *C*-termini of the polycationic isopeptides.

The mechanisms of the outstanding cell-penetrating activities of **1** and **2** still remain unclear, like canonical CPPs. However, the protein cargoes conjugated with **1** or **2** were certainly internalized into cells by direct penetration under high concentration conditions (50 μM). Recent studies have focused on rationally designed synthetic CPPs to obtain more efficient internalizations (direct penetrations or endosomal escapes) with low concentration treatments (~2 μM) and to confer resistance to proteolytic degradation[19–21,48–51]. Concomitantly, these attempts can reduce the amount of synthetic CPPs prepared by advanced chemical synthesis with high cost. In contrast, a cost-effective strategy allows for conducting protein intracellular deliveries under high concentration conditions (e.g., >50 μM) preferable for reliable direct penetration uptakes. The bacterial polycationic isopeptides, **1** and **2**, are intriguing new tools with this potential. Given the proof of these cell-penetrating activities, we propose the name *polycationic isopeptides entering cells (PIECEs)* for **1** and **2**. Although they are the first examples of bacterial PIECEs, further investigations of the cell-penetrating activities in bacterial polycationic isopeptides, such as γ-poly-L/D-diaminobutyric

acid[4,5], β-poly-L-diaminopropionic acid[6], ε-poly-L-β-lysine[7], cyanophycin[9,10], and multi-L-diaminopropionyl-poly-L-diaminopropionic acid (Supplementary Fig. 1a)[11], would expand the potential applications of the CPP-driven technology.

## Methods
**Chemicals**. Short-chain polyols, 2-propyn-1-ol, 3-buten-1-ol, tetraethylene glycol (PEG), triethylene glycol mono(2-propynyl) ether (PEG-alkyne), 11-azido-3,6,9-trioxaundecanol (PEG-azide), and 11-azido-3,6,9-trioxaundecan-1-amine (NH2-PEG-azide), were purchased from Tokyo Chemical Industry (Japan). β-hLys was purchased from Watanabe Chemical (Japan). DBCO-PEG4-5/6-FAM was purchased from Jena Bioscience (Germany). DBCO-Sulfo-N-hydroxysuccinimidyl ester (DBCO-Sulfo-NHS) was purchased from BROADPHARM (USA). FAM-NHS ester was purchased from ANASPEC (USA). R8-azide (**9**) and R8-FAM (**11**) were purchased from Eurofins Genomics (Japan). The endocytosis/macropinocytosis inhibitors, i.e., phenylarsine oxide (PAO), Dynasore, monodansylcadaverine (MDC), and ethylisopropyl amiloride (EIPA), were purchased from Fujifilm Wako Pure Chemical (Japan), Adipogen Life Sciences (USA), Sigma-Aldrich (USA), and TOCRIS bioscience (USA), respectively. All other chemicals used were of analytical grade.

**Bacterial strains, mammalian cell lines, and plasmids**. *S. albulus* NBRC14147 was used to produce ε-PαL and its ester derivatives. *E. coli* BL21(DE3) was used to overexpress mAG, mKO, and Cre recombinase. Human cervical cancer cells

(HeLa), human embryonic kidney cells (HEK293T), human hepatocellular carcinoma cells (HepG2), and human acute myelocytic leukemia cells (K562) were purchased from RIKEN BioResource Research Center (Japan). Human neuroblastoma cells (SH-SY5Y) and human dermal fibroblast cells (HDF) were purchased from the American Type Culture Collection (USA) and PromCell (Germany), respectively. The cell lines (HeLa, HEK293T, HepG2, K562) authenticated by RIKEN BioResource Research Center (Japan) were not further validated. The cell lines, SH-SY5Y and HDF, were not authenticated. We purchased the cell lines (HeLa, HEK293T, HepG2, K562) tested for mycoplasma contamination by RIKEN BioResource Research Center (Japan), but SH-SY5Y and HDF were not tested for mycoplasma contamination.

Two plasmids, pmAG1-S1 and pmKO1-MC1 (MBL, Japan), were used as the PCR templates to amplify the mAG and mKO genes, and the pmKO1-MC1 plasmid was also used to construct the Cre activity reporter plasmid. pKU470 (a gift from Dr. Haruo Ikeda)[52] was used as a PCR template to amplify the Cre recombinase gene.

**Microbiological production of clickable ε-PαL ester derivatives.** *S. albulus* NBRC14147 was grown in M3G medium[53] for 1 day at 28 °C. When the pH of the culture broth reached around 4, 0.2% (w/v) of the short-chain polyol (glycerol, 2-propyn-1-ol, 3-buten-1-ol, PEG, PEG-alkyne, or PEG-azide) was added to the culture medium. After culturing for 2 days at 28 °C, the resulting ε-PαL ester derivative in the culture broth was analyzed by HPLC-HR-ESI-MS (see below).

The ε-PαL ester derivatives were purified from the culture broth (1000 ml) using ion complex formation as described in our previous study[54], and then further purified by preparative HPLC using a reversed-phase column (Sunniest RP-AQUA, 5 μm, 10.0 × 250 mm; ChromaNik Technologies, Japan) at 35 °C at a flow rate of 7 ml min$^{-1}$ and with a linear gradient of acetonitrile in water in 0.1% (v/v) HFBA run over 30 min (20–50% [v/v] acetonitrile for 30 min). Fractions were collected and monitored by an ultraviolet (UV) detector at 210 nm. The fraction containing the purified ε-PαL ester derivatives was lyophilized to give a white powder (approximately 300 mg). The amounts of ε-PαL and the ε-PαL ester derivatives produced in the supernatant were determined according to the method reported by Itzhaki[55].

**Enzymatic synthesis of ε-PαL ester derivatives by Pls.** Using the method described in our previous study[56], the recombinant Pls (rPls) was purified from an overexpressing *Streptomyces* strain. We performed enzyme reactions with glycerol and PEG to determine whether the ε-PαL ester derivatives were produced by Pls. The reaction mixture containing 100 mM *N*-tris(hydroxymethyl)methyl-3-aminopropanesulfonic acid (TAPS)-NaOH (pH 8.5), 1 mM L-αLys, 5 mM MgCl₂, 5 mM ATP, 1 mM dithiothreitol (DTT), 20% (w/v) glycerol, 0.2% (w/v) NP-40, and 300 μg ml$^{-1}$ rPls was incubated with and without 1% (w/v) PEG at 25 °C for 10 min. The reaction mixture containing ε-PαL-glycerol (**3**) or ε-PαL-PEG (**6**) was then analyzed by HPLC-HR-ESI-MS (see below).

**Overexpressions and purifications of recombinant enzymes (mAG, mKO, and Cre recombinase).** The following three sets of PCR primers (Supplementary Table 29) were designed and used to amplify the mAG, mKO, and Cre recombinase genes: pET21a_mAG-F and pET21a_mAG-R for the construction of recombinant mAG; pET21a_mKO-F and pET21a_mKO-R for the construction of recombinant mKO; and pET21a_Cre-F and pET21a_Cre-R for the construction of recombinant Cre recombinase. The PCR products were ligated with the expression vector, pET21a (Novagen, USA). After confirming their DNA sequences, the resulting plasmids were introduced into *E. coli* BL21(DE3) for expression as *C*-terminally 6×His-tagged fusion proteins. The recombinant proteins (enzymes) were purified using standard protocols with nickel-nitriloacetic acid (Ni-NTA) Sepharose (Qiagen) and then used for further experiments.

**Synthesis of ε-PαL-cargo conjugates and R8-cargo conjugates.** ε-PαL-FAM (**10**): ε-PαL-PEG-azide (**8**, 25 μmol) was added to 50 ml of 80% (v/v) DMSO. Next, DBCO-PEG₄-5/6-FAM (27.5 μmol) was added and the mixture was stirred for 3 h at 25 °C. ε-PαL-FAM (**10**) in the reaction mixture was analyzed by HPLC-HR-ESI-MS and was then purified by preparative HPLC as described above. The fractions were collected and monitored by an ultraviolet (UV) detector at 254 nm. The fraction containing the purified **10** was lyophilized to give a yellow powder (approximately 50 mg).

ε-PαL-mAG (**12**), ε-PαL-mKO (**14**), and ε-PαL-Cre (**16**): The recombinant proteins (enzymes) (5 μmol) were added to 50 ml of 100 mM NaHCO₃, along with DBCO-Sulfo-NHS ester (10 μmol). After stirring for 4 h at 4 °C, the reaction mixtures were dialyzed with 20 mM sodium phosphate buffer (pH 6.0), and **8** (5.5 μmol) was added to 50 ml of the dialyzed solution and stirred for 4 h at 4 °C. Additionally, **12** and **14** were further purified by ion-exchange chromatography using a HiTrap SP HP column (Cytiva) on the AKTA system (GE Healthcare) after confirming the productions of **12**, **14**, and **16** by HPLC-HR-ESI-MS analysis (see below). Buffer A (20 mM sodium phosphate buffer, pH 6.0) and Buffer B (2 M NaCl and 20 mM sodium phosphate buffer, pH 6.0) were used as eluents. Flow rate: 3 ml min$^{-1}$; gradient: 0–20 column volume (CV) (B, 10–75%), 20–30 CV (B, 100%). Fractions were collected and monitored by a UV detector at 280 nm. The

fractions containing the purified proteins were concentrated by ultrafiltration (Vivaspin 6; Cytiva) and then used for further experiments. ε-PαL-Cre (**16**) was used for further experiments without the purification by ion-exchange chromatography.

R8-mAG (**13**) and R8-mKO (**15**): The recombinant mAG and mKO (0.5 μmol) were added to 5 ml of 100 mM NaHCO₃, along with DBCO-Sulfo-NHS ester (1 μmol). After stirring for 4 h at 4 °C, the reaction mixtures were dialyzed with 20 mM sodium phosphate buffer (pH6.0), and R8-azide (**9**, 0.55 μmol) was added to 5 ml of the dialyzed solution and stirred for 4 h at 4 °C. The resulting conjugates, **13** and **15**, were further purified by ion-exchange chromatography using a HiTrap SP HP column (Cytiva) on the AKTA system (GE Healthcare) as described above, after confirming the productions of **13** and **15** by HPLC-HR-ESI-MS analysis (see below). The fractions containing **13** and **15** were concentrated by ultrafiltration (Vivaspin 6) and then used for further experiments.

**Construction of the Cre activity reporter plasmid.** The synthetic DNA fragment carrying two loxP sequences, one TGA stop codon, and the ampicillin resistance gene (Amp$^r$ gene) (Supplementary Table 30) was ligated with pmKO1-MC1 digested by *Bam* HI and *Eco* RI. The resulting plasmid, pmKO_loxP, was digested with *Eco* RI and *Hind* III and ligated with the mAG gene fragment amplified by the PCR primers (pmKO1_mAG-F and pmKO1_mAG-R, Supplementary Table 29) to yield the Cre activity reporter plasmid (pmKO_loxP_mAG).

**Synthesis of ε-PαL-mAb$^{FAM}$ (17).** Anti-α-tubulin antibody (mAb, 1 μmol) (Fujifilm Wako Pure Chemical, Japan) was added to 10 ml of 100 mM NaHCO₃, along with FAM-NHS ester (4 μmol). After stirring for 37 °C for 30 min, DBCO-Sulfo-NHS ester (4 μmol) was added to the reaction mixture and stirred at 37 °C for 30 min. The reaction mixtures were dialyzed with 20 mM sodium phosphate buffer (pH 6.0), and **8** (2 μmol) was added to 10 ml of the dialyzed solution and stirred for 4 h at 4 °C. Further, **17** in the reaction mixture was concentrated by ultrafiltration (Vivaspin 6) and used for further experiments after confirming the production of **17** by HPLC-HR-ESI-MS analysis.

**Mammalian cell culture.** Cell lines were grown at 37 °C in a humidified atmosphere with 5% CO₂. The cell lines and corresponding media were as follows: HeLa, Dulbecco's modified Eagles Medium (DMEM) with 10% fetal bovine serum (FBS); HEK293T, DMEM with 10% FBS; HepG2, DMEM with 10% FBS; HDF, DMEM with 10% FBS; SH-SY5Y, DMEM/Ham's F-12 with 10% FBS; and K562, RPMI1640 with 10% FBS. All media contain 100 U/ml penicillin and 100 μg/ml streptomycin.

**CM observation of cytosolic deliveries.** Cells were seeded into each well of an 8-well chamber slide glass (Matsunami, Japan) and were incubated at 37 °C for 24 h. A fresh medium containing a sample (**10–17**, **24**, or **25**) was added to the cells and then incubated at 4 °C or 37 °C for 60 min or 120 min after washing the cells with the medium. The cells were then washed with PBS, 100 μg ml$^{-1}$ heparin in PBS, and then two more times with PBS. We added 4% paraformaldehyde in PBS to fix the cells, and incubated them at 25 °C for 30 min. 5 μg ml$^{-1}$ wheat germ agglutinin (WGA)/Alexa594 (or Alexa488) in PBS was added and incubated at 25 °C for 10 min to stain the cell membrane after washing the cells thrice with PBS. Then, the cells were treated with ProLong$^{TM}$ Gold antifade reagent with DAPI (Invitrogen, USA). The cells were observed by using an LSM 900 Confocal Laser Scanning Microscope (ZEISS).

**Quantification of cellular uptake.** HeLa cells were seeded into each well of a 12-well plate (Falcon) and were incubated for 24 h at 37 °C in DMEM containing 10% FBS. Next, a fresh medium containing a sample (**10–12**, **24**, or **25**) was added to the cells and then incubated at 4 °C or 37 °C for 5–60 min. The cells were then washed with PBS, followed by 100 μg ml$^{-1}$ heparin in PBS, and then twice with PBS. 300 μl of lysis buffer (1% Triton X-100 in 100 mM Tris-HCl, pH 8.0) was added to the cells for cell lysis. The resulting cell lysates were centrifuged for 30 min (21,500 × *g* at 4 °C), and the fluorescence (485 ex/535 em) of the supernatants was measured with a fluorescent plate reader (SpectraMax iD3; Molecular Devices or SYNERGY 4; BioTek).

In the assays using endocytosis/macropinocytosis inhibitors, 1 μM PAO, 100 μM MDC, 100 μM Dynasore, or 100 μM EIPA was added to the cell culture.

**Time-lapse imaging of ε-PαL-mAG (12) and R8-mKO (15) during their cellular uptakes.** HeLa cells were seeded in a 35 mm glass-bottom dish (Matsunami, Japan) and were incubated in DMEM containing 10% FBS for 24 h at 37 °C. Next, a fresh warmed (37 °C) medium containing 50 μM of ε-PαL-mAG (**12**) or R8-mKO (**15**) was added, and the time-lapse images of the cells were captured every 1 min for 30 min by an LSM 900 Confocal Laser Scanning Microscope (ZEISS).

**Time-lapse imaging to track ε-PαL-mAG (12) and R8-mKO (15) after their cellular uptakes.** HeLa cells were seeded in a 35 mm glass-bottom dish (Matsunami, Japan) and were incubated at 37 °C in DMEM containing 10% FBS for 24 h. Next, a fresh medium containing 50 μM of ε-PαL-mAG (**12**) or R8-mKO (**15**) was added, and the cells were incubated at 37 °C for 60 min. The cells were then washed

with PBS, followed by 100 µg ml$^{-1}$ heparin in PBS, and then two more times with PBS. Finally, fresh DMEM containing 10% FBS was added and incubated at 37 °C. The time-lapse images were captured every 10 min for 12 h by optical sectioning microscopy (All-in-One Fluorescence Microscope BZ-X810; KEYENCE) to track **12** or **15** after the cellular uptakes.

**Cre/loxP recombination mediated by the intracellularly delivered ε-PαL-Cre (16)**. HEK293T cells were transfected with the plasmid pmKO_loxP_mAG using FuGENE Transfection (Promega) according to the manufacturer's specifications. The cells were washed with DMEM medium and then treated with 50 µM of ε-PαL-Cre (**16**) at 37 °C for 60 min. The cells were then washed with PBS, followed by 100 µg ml$^{-1}$ heparin in PBS, and then two more times with PBS. Finally, fresh DMEM containing 10% FBS was added and incubated at 37 °C for 24 h to express a fluorescent protein. The live-cell imaging was captured by fluorescence microscopy (Leica DMi8).

**Synthesis of acceptor substrates (18, 20, and 22) for ORF19**. N,N-bis(tert-butoxycarbonyl)-L-β-Lysine [(Boc)$_2$-L-βLys; 1.5 mmol]·DCHA, which had been synthesized as described in the Supplementary Note according to the methods previously described[57], was acid extracted with AcOEt/10% citric acid to remove DCHA. NH$_2$-PEG-azide (0.55 mmol) and DMT-MM (1.5 mmol) were added to an MeOH solution (10 ml) of the free (Boc)$_2$-L-βLys, and the mixture was stirred at room temperature overnight. The residue was dissolved into Et$_2$O (10 ml), washed with brine, and dried (Na$_2$SO$_4$) after a concentration in vacuo. A crude (Boc)$_2$-L-βLys-PEG-azide thus obtained by evaporation was added to 4N HCl/AcOEt (20 ml), and the mixture was stirred at room temperature for 3 h. After evaporation, HCl remained in the mixture was further removed by azeotropic evaporation with toluene/MeOH. The residue was dissolved in water and lyophilized, affording L-βLys-PEG-azide (**18**)·HCl as a colorless oil (22% yield, 2 steps). L-βhLys-PEG-azide (**20**)·HCl (38% yield) and L-αLys-PEG-azide (**22**)·HCl (81% yield) were synthesized from L-βhLys and L-αLys, respectively, by bis-Boc derivatization, condensation with NH$_2$-PEG-azide, and subsequent deprotection. Their NMR data are summarized in Supplementary Tables 19, 21, and 23.

**ORF19 enzyme reaction**. According to the method described in our previous study[3], the recombinant ORF19 (rORF19) was purified from an overexpressing E. coli strain. A reaction mixture (100 µl) consisting of 50 mM TAPS (pH 9.0), 18% (v/v) glycerol, 5 mM ATP, 5 mM MgCl$_2$, 5 mM L-βLys, 1 mM acceptor substrate (**18**, **20**, or **22**), and 0.1 mg ml$^{-1}$ rORF19 was incubated for 1 h at 30 °C. The reaction was quenched by heating at 100 °C for 5 min, the mixture was centrifuged, and the supernatant was analyzed by HPLC-HR-ESI-MS (see below).

**Chemoenzymatic synthesis of ε-OβL$^m$-PEG-azide (21)**. A reaction mixture (20 ml) consisting of 50 mM TAPS (pH 9.0), 18% (v/v) glycerol, 5 mM ATP, 5 mM MgCl$_2$, 5 mM L-βLys, 1 mM **20**, and 1 mg ml$^{-1}$ rORF19 was incubated at 30 °C overnight (1st round enzyme reaction, Supplementary Table 25). The resulting ε-OβL$^m$-PEG-azide (**21**) was purified by a disposable ion-exchange column (InertSep MC-2 column; GL Science, Japan). The mixture (20 ml) was added to 80 ml of 20% (v/v) aqueous methanol and applied to a column of InertSep MC-2 (Na$^+$ form; 1 g: equilibrated with 20% (v/v) aqueous methanol) after adjusting the pH of the reaction mixture to 7 by HCl. The product (short-chain **21**, 2–9 mer) was eluted with 10 ml of 1 M NH$_4$OH after the column had been washed with 50 ml of 20% (v/v) aqueous methanol. The elution fraction was lyophilized and then used as an acceptor substrate for the 2nd round enzyme reaction. The 2nd round enzyme reaction mixture (20 ml) consisting of 50 mM TAPS (pH 9.0), 18% (v/v) glycerol, 5 mM ATP, 5 mM MgCl$_2$, 5 mM L-βLys, 1 mM short-chain **21** (2–9 mer), and 1 mg ml$^{-1}$ rORF19 was incubated overnight at 30 °C (Supplementary Table 25). The reaction product, long-chain **21** (2–13 mer), was purified by InertSep MC-2 and then further purified by preparative HPLC using a reversed-phase column (Sunniest RP-AQUA: 5 µm, 10.0 × 250 mm; ChromaNik Technologies, Japan) at 40 °C at a flow rate of 7 ml min$^{-1}$ and with a linear gradient of acetonitrile in water in 0.1% (v/v) HFBA run over 36 min [20–38% (v/v) acetonitrile for 36 min]. Fractions were collected and monitored by an ultraviolet (UV) detector at 210 nm. The fraction containing the purified **21** was lyophilized and used for further experiments.

**Synthesis of ε-OβL$^m$-FAM (24) and ε-OβL$^m$-mAG (25)**. ε-OβL$^m$-FAM (**24**) and ε-OβL$^m$-mAG (**25**) were synthesized by the same procedure as used for the synthesis of **10** and **12** (see above).

**HPLC-HR-ESI-MS analysis**. ε-PαL (**1**), ε-PαL-glycerol (**3**), ε-PαL-alkyne (**4**), ε-PαL-alkene (**5**), ε-PαL-PEG (**6**), ε-PαL-PEG-alkyne (**7**), ε-PαL-PEG-azide (**8**), and ε-PαL-FAM (**10**) were analyzed by HPLC-HR-ESI-MS (maXis plus; Bruker) using a reversed-phase column (SunShell HFC18–16: 2.6 µm, 2.1 × 150 mm; ChromaNik Technologies, Japan) at 40 °C at a flow rate of 0.3 ml min$^{-1}$ and with a three-step linear gradient of acetonitrile (ACN) in water in 0.1% (v/v) heptafluorobutyric acid (HFBA) (Fujifilm Wako Pure Chemical, Japan) run over 30 min [10–25% (v/v)

ACN for 5 min, 25–60% (v/v) ACN for 17 min, 60–95% (v/v) ACN for 5 min, and 95–95% (v/v) ACN for 3 min].

ε-PαL-mAG (**12**), R8-mAG (**13**), ε-PαL-mKO (**14**), R8-mKO (**15**), and ε-PαL-Cre (**16**) were analyzed by HPLC-HR-ESI-MS (maXis plus) using a reversed-phase column (Sunshell C8-30HT: 3.4 µm, 2.1 × 150 mm; ChromaNik Technologies) at 70 °C at a flow rate of 0.3 ml min$^{-1}$ and with a linear gradient of ACN in water in 0.1% (v/v) trifluoroacetic acid (TFA) run over 30 min [**12** and **13**, 31–37% (v/v) ACN for 30 min; **14** and **15**, 2–90% (v/v) ACN for 30 min; **16**, 20–70% (v/v) ACN for 30 min].

L-βLys-PEG-azide (**18**), L-βhLys-PEG-azide (**20**), and L-αLys-PEG-azide (**22**) were analyzed by HPLC-HR-ESI-MS (maXis plus) using a reversed-phase column (SunShell HFC18-16: 2.6 µm, 2.1 × 150 mm; ChromaNik Technologies, Japan) at 40 °C at a flow rate of 0.3 ml min$^{-1}$ and with a three-step linear gradient of ACN in water in 0.1% (v/v) HFBA run over 30 min (5–25% [v/v] ACN for 5 min, 25–35% [v/v] ACN for 17 min, 35–95% [v/v] ACN for 5 min, and 95–95% [v/v] ACN for 3 min].

**NMR spectroscopy**. $^1$H- and $^{13}$C-NMR spectra were recorded at 600 and 150 MHz, respectively, using a JNM-ECA 600 NMR (JEOL). One- and two-dimensional experiments (double quantum Filtered-correlation spectroscopy [DQF-COSY] and constant time-heteronuclear multi-bond connectivity [CT-HMBC]) were performed at ambient temperature. The samples were dissolved in D$_2$O.

**Statistics and reproducibility**. All statistical analyses were performed using GraphPad Prism software (ver. 8; GraphPad, San Diego, CA, USA). Differences were considered significant when the calculated p-value was < 0.05. No statical methods were used to predetermine sample size. Where statical analyses were performed, the sample sizes are listed in the corresponding figure legends. All experiments presented in the manuscript were reproducible. The results are presented as the mean ± standard deviation (s.d.) (n = 3). Information regarding number of biological and technical replicates is reported for individual experiments in the figure legends.

**Reporting summary**. Further information on research design is available in the Nature Research Reporting Summary linked to this article.

## Data availability
All data is available in the main Figures and Supplementary Figures. Raw data for Figs. 3c–j, 4b–g, 8b–e, Supplementary Fig. 7a–f, and Supplementary Table 19 are available in the Supplementary Data 1–5. The plasmid, pmKO_loxP_mAG, constructed in this study has been deposited in the Addgene under the ID number 192857.

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

## Acknowledgements

This research was supported by a JSPS KAKENHI grant for Scientific Research on Innovative Areas 16H06445 (Y.H.), by the JSPS A3 Foresight Program, by JSPS KAKENHI grants16H03284 (Y.H.) and 20H02918 (Y.H.), and by the Japan Foundation for Applied Enzymology (Y.H.), the Nagase Science and Technology Foundation (Y.H.), and the Amano Enzyme Foundation (Y.H.). We thank H. Fujino for discussion on the cell culture.

## Author contributions

Y.H. conceived and designed the project. Y.T., K.U., and K.Y. performed experiments and collected data regarding ε-PαL. H.K. developed protocols for purification of ε-PαL ester derivatives. K.K. and C.M. performed experiments and collected data regarding ε-OβL. T.I. discussed and interpreted results from cell cultures. C.M., Y.O., and T.D. analyzed the compounds and elucidated their chemical structures. Y.K. synthesized L-βLys. Y.H. wrote the manuscript, and all other authors reviewed and commented on it.

## Competing interests

Y.H. and K.U. are co-inventors on a patent application (PCT/JP2018/031153) filed by Fukui Prefectural University relating to work in this manuscript. Y.H. and C.M. are co-founders of MicrobeChem Inc. The remaining authors declare no competing interests.
