## [Peer Review File · Communications Biology]

Reviewers' comments:

Reviewer #1 (Remarks to the Author):

The manuscript #COMMSBIO-22-1850-T entitled "First direct evidence for direct cell-membrane penetrations of polycationic homopoly(amino acid)s produced by bacteria" by Takeuchi et al. report the experimental proof of cell-penetrating properties of polycation homopoly(amino acids)s, e.g., ϵ -poly-L- α -lysine (ϵ -PaL) and ϵ -oligo-L- β -lysine (ϵ -O β L) and delivery of antibody covalently conjugated using click chemistry in the cells. The authors used a testable hypothesis and successfully proved the CPPs properties using confocal microscopy, enzymatic activity experiments, and cellular uptake studies. The videos and high resolutions figures clearly demonstrate the uptake of conjugates and polycationic peptides as compared to classical polyarginine (R8 and R12) peptides. Furthermore, the authors used appropriate control experiments using various inhibitors and temperatures to provide fundamental observation for the mechanism of cellular uptake of ϵ -PaL and ϵ -O β L. The chemical characterization of isopeptides, conjugates, and antibodies were fully elaborated and provided. The manuscript is well prepared and written except for the cytotoxicity of these polycationic isopeptides. The reported finding greatly influences ongoing work for targeting cytoplasmic protein targets and their translation for therapeutic development. Therefore, the manuscript has a higher impact on the field and is recommended for publication with a suggestion to improve the manuscript with the discussion related to cytotoxicity.

Reviewer #2 (Remarks to the Author):

Summary

The authors report the modification and characterization of naturally occurring iso-peptides for the delivery of fluorophores, protein, and antibody cargos into cells, termed polycationic isopeptides entering cells (PIECES). The study identifies a series of ϵ -poly-L- α -lysine (ϵ -PaL) and ϵ -oligo-L- β -lysine (ϵ -O β L) isopeptides conjugated to synthetic fluorophores, fluorescence proteins, Cre enzyme and a monoclonal antibody and evaluates cellular uptake as well as uptake mechanism of uptake into the cytoplasm and nucleus of HeLa cells.

Significance

The intracellular delivery of biomacromolecular agents, such as enzymes and antibodies, is a major challenge in biomedical research. The manuscript builds on previous work by the authors on homopoly(amino acid)s and homooligo(amino acid)s (Hamano, Y., et al. Nat. Prod. Rep. 30, 1087-97 (2013); Yamanaka, K., et al. Nat. Chem. Biol. 4, 766-72 (2008)) by using these to deliver cargos into cells. Given the importance and promise of intracellular delivery, this manuscript represents an important and timely contribution. The authors present an important alternative chemotype to standard (eupeptides) cell penetrating peptides (CPPs) and present a convincing argument that these can be used to transport cargos into cells. The work cites ease of production and enhances stability as major advantages of PIECES over standard CPPs but leaves open several important questions that would allow further comparison of the two approaches.

Major Comments

The work presented in the manuscript is technical sound and well executed. The experimental methods are well chosen and described in sufficient detail for reproduction; novel compounds and biologics are thoroughly characterized. However, some aspects of this work would benefit from clearer presentation and require further discussion:

- A major omission in this study is toxicity data; – especially given that the use of fixed cell microscopy does not allow examination of cellular integrity. Addition of some toxicity data by PI or MTT for lead compounds would strengthen the claim that these agents can transport cargo into cells without major disruption.
- Although 50 μM is used consistently throughout the manuscript for all (ϵ -PaL) and (ϵ -O β L), as well as conjugate cargos, antibody uptake studies were conducted at 10 μM . Is there a reason for this? It seems counterintuitive that the largest cargo can be delivered at lower concentrations, especially given that only about one third of IgG is decorated with ϵ -PaL (Supp Table 18).
- The manuscript leaves open the questions of stability and efficacy which are crucial when comparing this technology to current state of the art approaches. Do PIECEs provide improved stability over traditional, especially cyclised CPPs? Is a reported efficacy of 50 μM suitable for application in biomedical research or drug development, especially in light of recent studies reporting low micromolar efficacy (Nat. Chem., 2022, 14:274; Nat. Chem., 2022, 14:284; Angew. Chem. Int. Ed., 2021, 60: 19804). Given the scope of this communication, addressing these two points in the discussion might be most appropriate.

Minor Comments

- I suggest individual numbering of compound 21 and 24 separated ϵ -O β L oligomers with individual compound numbers as these are separate chemical entities and the current nomenclature is a bit confusing.
- Compound 25 looks like it is a mixture of two species, one single ϵ -O β L conjugate and one that carries two ϵ -O β L (signal > 29 kDa).

A few sections of the manuscript are unclear and potentially misleading, ex:

- Pg.5, ln 2 "STs with molecular weights over 1,000 (ST-B, ST-A, and ST-X) (Fig.1b) should be impermeable." – Are they impermeable?
- Pg.5, ln 19 "However, there have been few successes by a chemical approach." – There have been quite a few successful chemical approaches to solve this issue.
- Pg.8, ln 7 "was chemically derivatized with a DBCO group" – detail on the type of chemistry used to accomplish this and molar ratios should be added here.

...

Recommendation

This work is technically well executed, comprehensive and of interest in the field intracellular delivery. I recommend publication of this manuscript in Nature Communications Biology provided that the concerns raised above are addressed.

Reviewer #3 (Remarks to the Author):

The authors found that bacteria-derived polylysine isopeptides 1 and 2 as a new type of CPPs. They showed that the isopeptides are able to deliver a fluorescent dye and proteins of different sizes into HeLa cells. Preliminary experiments suggest that they enter cells by endocytosis at low concentrations and direct translocation at high concentrations, just like conventional cationic CPPs. The new CPPs have the advantage of much greater proteolytic stability and may be potentially useful as research tools and drug delivery agents. The manuscript needs some modifications as detailed below.

- 1) The authors fixed the cells before confocal imaging, which is not ideal. Live-cell imaging is doable and preferred, as fixation has been reported to change the intracellular distribution of CPPs.
- 2) The conclusions on entry mechanism (endocytosis vs direct translocation) for some of the peptides (e.g., 15 and 17) are not supported by conclusive data. The statement that "15 should never escape

the endosomes" is not true. The confocal images cannot detect low levels of the peptide inside the cytosol. Likewise, 17 may enter the cell by both endocytosis and direct translocation, but endocytosis dominates under the experimental conditions.

3) The authors claim isopeptides 1 and 2 as "highly effective" for protein delivery, which is not supported by data. First, delivery of proteins requires 50 μM concentration, which can hardly be considered as "highly effective". Second, most of the data are from confocal microscopy which is not quantitative. The only functional delivery is with Cre, which requires very low concentration to have an effect.

4) On page 6 line 19: What does "most potent clickable" mean?

Response to reviewers

Reviewer #1:

Thank you so much for your valuable comments concerning our original manuscript.

Your comment #1:

The manuscript is well prepared and written except for the cytotoxicity of these polycationic isopeptides.

Answer:

As we mentioned in the Introduction section (page 4, line 14 in the revised manuscript), ϵ -poly-L- α -lysine (ϵ -P α L) shows potent antimicrobial activities and nondetectable levels of cytotoxicity. The nontoxicity of ϵ -P α L has also been proved by the fact that ϵ -P α L is used as a natural food preservative in several countries. However, we did not have enough data to address your concern about *in vitro* cytotoxicities (same concern as Reviewer #2). We therefore performed additional experiments to confirm the cytotoxicities against the cell lines used in the ϵ -P α L uptake studies. Expectedly, ϵ -P α L showed no cytotoxicities in four cell lines (K562, HeLa, HepG2, and HEK293T). This additional data was added in the Result section and shown in Supplementary Table 19 in the revised manuscript (page 10, line 21 – page 11, line 3 in the revised manuscript).

In ϵ -oligo-L- β -lysine (ϵ -O β L), we previously reported that it has no cytotoxicity in HeLa cells (Ref. 3). The cytotoxicities of ϵ -P α L and ϵ -O β L are also emphasized in the Discussion section of the revised manuscript (page 15, line 4 in the revised manuscript).

Reviewer #2:

Thank you so much for your valuable comments concerning our original manuscript. According to your suggestion, we have revised the manuscript carefully.

Your comment #1 (Major Comment):

A major omission in this study is toxicity data; – especially given that the use of fixed cell microscopy does not allow examination of cellular integrity. Addition of some toxicity data by PI or MTT for lead compounds would strengthen the claim that these agents can transport cargo into cells without major disruption.

Answer:

As we mentioned in the Introduction section (page 4, line 14 in the revised manuscript), ϵ -poly-L- α -lysine (ϵ -P α L) shows potent antimicrobial activities and nondetectable levels of cytotoxicity. The nontoxicity of ϵ -P α L has also been proved by the fact that ϵ -P α L is used as a natural food preservative in several countries. However, we did not have enough data to address your concern about *in vitro* cytotoxicities (same concern as Reviewer #1). We therefore performed additional experiments to confirm the cytotoxicities against the cell lines used in the ϵ -P α L uptake studies. Expectedly, ϵ -P α L showed no cytotoxicities in four cell lines (K562, HeLa, HepG2, and HEK293T). This additional data was added in the Result section and shown in Supplementary Table 19 in the revised manuscript (page 10, line 21 – page 11, line 3 in the revised manuscript).

In ϵ -oligo-L- β -lysine (ϵ -O β L), we previously reported that it has no cytotoxicity in HeLa cells (Ref. 3). The cytotoxicities of ϵ -P α L and ϵ -O β L are also emphasized in the Discussion section of the revised manuscript (page 15, line 4 in the revised manuscript).

Your comment #2 (Major Comment):

Although 50 μ M is used consistently throughout the manuscript for all (ϵ -P α L) and (ϵ -O β L), as well as conjugate cargos, antibody uptake studies were conducted at 10 μ M. Is there a reason for this? It seems counterintuitive that the largest cargo can be delivered at lower concentrations, especially given that only about one third of IgG is decorated with ϵ -P α L (Supp Table 18).

Answer:

We employed the fluorescently labeled anti- α -tubulin antibody in this experiment to examine if we could detect a fluorescent image of the microfilamentous cytoskeleton. We speculated that a high concentration (50 μ M) could not give us a distinguishable image (cytoskeleton or diffusing signals) due to the excess and α -tubulin-unbound antibody that is unremovable in the living cells. In contrast, the low concentration (e.g., 2 μ M) would result in only punctate signals suggestive of endosomes. We therefore decided to employ 10 μ M in this experiment. The revised manuscript was improved by adding these explanations (page 10, line 7-12).

Your comment #3 (Major Comment):

The manuscript leaves open the questions of stability and efficacy which are crucial when comparing this technology to current state of the art approaches. Do PIECEs provide improved stability over traditional, especially cyclised CPPs? Is a reported efficacy of 50 μ M suitable for application in biomedical research or drug development, especially in light of recent studies reporting low micromolar efficacy (Nat. Chem., 2022, 14:274; Nat. Chem., 2022, 14:284; Angew. Chem. Int. Ed., 2021, 60: 19804). Given the scope of this communication, addressing these two

points in the discussion might be most appropriate.

Answer:

It is tough to know the difference in the stabilities between bacterial PIECEs (linear isopeptides) and the recent synthetic CPPs because it depends on the activities of peptidases and proteinases. The cyclic CPPs have great potential for overcoming the critical degradation problem and delivering cargoes via direct penetration at low concentrations. However, these recent approaches require advanced chemical synthesis. Therefore, many biologists would hesitate to pursue them, and their higher cost remains a critical issue to be solved. In contrast, the cost-effective strategy using the bacterial PIECEs allows us to perform protein intracellular deliveries under high concentration conditions (e.g. > 50 μ M) preferable for reliable direct penetration uptakes.

The bacterial PIECEs are thus intriguing new tools with the potential to expand the application of CPP-driven technologies. This strong point was discussed in the revised manuscript (page 15, line 7-17).

Your comment #4 (Minor Comment):

I suggest individual numbering of compound 21 and 24 separated ϵ -O β L oligomers with individual compound numbers as these are separate chemical entities and the current nomenclature is a bit confusing.

Answer:

We improved the manuscript (page 12, line 20-23)

Your comment #5 (Minor Comment):

Compound 25 looks like it is a mixture of two species, one single ϵ -O β L conjugate and one that carries two ϵ -O β L (signal > 29 kDa)

Answer:

Yes, you are correct. The figure in Supplementary table 28 was revised to clarify this.

Your comment #6 (Minor Comment):

Pg.5, ln 2 “STs with molecular weights over 1,000 (ST-B, ST-A, and ST-X) (Fig.1b) should be impermeable.” – Are they impermeable?

Answer:

The sentence was corrected (Page 5, line 1-2).

Your comment #7 (Minor Comment):

Pg.5, ln 19 “However, there have been few successes by a chemical approach.” – There have been quite a few successful chemical approaches to solve this issue.

Answer:

The sentence was revised as you suggested (page 5, line 18-19).

Your comment #8 (Minor Comment):

Pg.8, ln 7 “was chemically derivatized with a DBCO group” – detail on the type of chemistry used to accomplish this and molar ratios should be added here.

Answer:

The sentence was revised as you suggested (page 8, line 6-7).

Reviewer #3:

Thank you so much for your valuable comments concerning our original manuscript. According to your suggestion, we have revised the manuscript carefully.

Your comment #1:

The authors fixed the cells before confocal imaging, which is not ideal. Live-cell imaging is doable and preferred, as fixation has been reported to change the intracellular distribution of CPPs.

Answer:

To confirm the intracellular uptake routes in ϵ -P α L-mAG (12) and R8-mKO (15), we performed time-lapse imaging with confocal microscopy (Please see page 8 line 19 – page 9 line 19 and Supplementary Video 1 - 4).

Your comment #2:

The conclusions on entry mechanism (endocytosis vs direct translocation) for some of the peptides (e.g., 15 and 17) are not supported by conclusive data. The statement that "15 should never escape the endosomes" is not true. The confocal images cannot detect low levels of the peptide inside the cytosol. Likewise, 17 may enter the cell by both endocytosis and direct

translocation, but endocytosis dominates under the experimental conditions.

Answer:

We appreciate this suggestion for improving clarity. We improved the sentences you pointed out in the revision (page 9, line 18-19; page 10, line 13; page 13, line 15-16).

Your comment #3:

The authors claim isopeptides 1 and 2 as "highly effective" for protein delivery, which is not supported by data. First, delivery of proteins requires 50 uM concentration, which can hardly be considered as "highly effective". Second, most of the data are from confocal microscopy which is not quantitative. The only functional delivery is with Cre, which requires very low concentration to have an effect.

Answer:

We removed "highly" in the revision (page 14, line 3-4).

Recent approaches using CPPs require advanced chemical synthesis. Therefore, many biologists would hesitate to pursue them, and their higher cost remains a critical issue to be solved. In contrast, the cost-effective strategy using the bacterial PIECEs (1 and 2) allows us to perform protein intracellular deliveries under high concentration conditions (e.g. > 50 μM) preferable for reliable direct penetration uptakes.

The bacterial PIECEs are thus intriguing new tools with the potential to expand the application of CPP-driven technologies. Based on these our ideas, we improved the Discussion section in the revised manuscript (page 15, line 7-17).

Your comment #4:

On page 6 line 19: What does "most potent clickable" mean?

Answer:

The sentence was revised by removing "most potent clickable" (page 6, line 19-21).

REVIEWERS' COMMENTS:

Reviewer #1 (Remarks to the Author):

The authors has provided appropriate response for my comments. I have no more questions or suggestion.

Reviewer #2 (Remarks to the Author):

Thank you for the careful revision of the manuscript. The additional cytotoxicity data and discussion points are convincing and I support the publication of the revised article in Nature Communications Biology.

Reviewer #3 (Remarks to the Author):

The revision was satisfactory.

Response to reviewers

Reviewer #1:

Thank you so much for your valuable comments concerning our original manuscript.

Your comment #1:

The authors has provided appropriate response for my comments. I have no more questions or suggestion.

Answer:

Thank you.

Reviewer #2:

Thank you so much for your valuable comments concerning our original manuscript.

Your comment #1 (Major Comment):

Thank you for the careful revision of the manuscript. The additional cytotoxicity data and discussion points are convincing and I support the publication of the revised article in Nature Communications Biology.

Answer:

Thank you.

Reviewer #3:

Thank you so much for your valuable comments concerning our original manuscript.

Your comment #1:

The revision was satisfactory.

Answer:

Thank you.